# Sex Differences in Plasma MicroRNA Biomarkers of Early and Complicated Diabetes Mellitus in Israeli Arab and Jewish Patients

**DOI:** 10.3390/ncrna5020032

**Published:** 2019-04-05

**Authors:** Ari Meerson, Azwar Najjar, Elias Saad, Wisam Sbeit, Masad Barhoum, Nimer Assy

**Affiliations:** 1MIGAL Galilee Research Institute, Kiryat Shmona 1101602, Israel; 2Tel Hai Academic College, Upper Galilee 1220800, Israel; 3Department of Internal Medicine A, Galilee Medical Center, Nahariya 89, Israel; dr_azwarnajjar@yahoo.com (A.N.); eliass@gmc.gov.il (E.S.); NimerA@gmc.gov.il (N.A.); 4Department of Gastroenterology, Galilee Medical Center, Nahariya 89, Israel; wisams@gmc.gov.il; 5Director, Galilee Medical Center, Nahariya 89, Israel; masad.barhoum@naharia.health.gov.il; 6The Azrieli Faculty of Medicine, Bar Ilan University, Safed 5290002, Israel

**Keywords:** microRNAs, extracellular, blood-borne, biomarkers, diabetes mellitus, complications, sex differences

## Abstract

MicroRNAs play functional roles in the etiology of type 2 diabetes mellitus (T2DM) and complications, and extracellular microRNAs have attracted interest as potential biomarkers of these conditions. We aimed to identify a set of plasma microRNAs, which could serve as biomarkers of T2DM and complications in a mixed Israeli Arab/Jewish patient sample. Subjects included 30 healthy volunteers, 29 early-stage T2DM patients, and 29 late-stage T2DM patients with renal and/or vascular complications. RNA was isolated from plasma, and the levels of 12 candidate microRNAs were measured by quantitative reverse transcription and polymerase chain reaction (qRT-PCR). MicroRNA levels were compared between the groups and correlated to clinical measurements, followed by stepwise regression analysis and discriminant analysis. Plasma miR-486-3p and miR-423 were respectively up- and down-regulated in T2DM patients compared to healthy controls. MiR-28-3p and miR-423 were up-regulated in patients with complicated T2DM compared to early T2DM, while miR-486-3p was down-regulated. Combined, four microRNAs (miR-146a-5p, miR-16-2-3p, miR-126-5p, and miR-30d) could distinguish early from complicated T2DM with 77% accuracy and 79% sensitivity. In male patients only, the same microRNAs, with the addition of miR-423, could distinguish early from complicated T2DM with 83.3% accuracy. Furthermore, plasma microRNA levels showed significant correlations with clinical measurements, and these differed between men and women. Additionally, miR-183-5p levels differed significantly between the ethnic groups. Our study identified a panel of specific plasma microRNAs which can serve as biomarkers of T2DM and its complications and emphasizes the importance of sex differences in their clinical application.

## 1. Introduction

Obesity, the associated type 2 diabetes mellitus (T2DM), and the complications of T2DM are leading health problems in developed countries. Type 2 diabetes mellitus is a metabolic disorder characterized by hyperglycemia, which arises from insufficient pancreatic insulin secretion, insulin resistance in peripheral tissues, and inadequate suppression of glucagon production. Complications of T2DM include vascular damage, blindness, and kidney failure. 

MicroRNAs (miRNAs) are short (20–27 nucleotide) endogenous RNA molecules which associate with argonaute proteins to regulate messenger RNA (mRNA) stability and translation [1]. The role of miRNAs in various biological processes, including adipocyte differentiation, metabolic integration, insulin resistance and appetite regulation [2], and the deregulation of many miRNAs in metabolic tissues of obese animals and humans [2,3,4,5], have been characterized. Our own study, among others, showed that miR-221 is up-regulated in the adipose tissue of obese subjects and regulates a cellular metabolism-related protein network, and specifically the adiponectin receptor ADIPOR1, which affects insulin sensitivity [4,6]. MicroRNAs are also functionally involved in the complications of T2DM, such as nephropathy [7,8] and cardiovascular complications [9,10].

MicroRNAs can be remarkably stable in biofluids, and specifically plasma/serum [11,12]. This stability appears to be conferred by the binding of proteins, or encapsulation in lipid vesicles, of the miRNA molecules [13]. The specific profiles of these extracellular miRNAs in biofluids and the particular changes in these profiles under the conditions of disease suggest the potential use of these RNA species as biomarkers useful for diagnosis and stratification of diseases, including metabolic disease [14,15]. The use of human plasma/serum samples for detection of miRNAs can either help to detect early onset of various diseases or provide early clues to the response of patients to drugs. By now, dozens of studies have described subsets of miRNAs which show differential expression in the circulation of T2DM patients; however, meta-analyses show that relatively few miRNA-based biomarkers are informative across multiple studies and/or populations [16,17]. These findings encourage further investigation into the potential clinical use of miRNAs as biomarkers of T2DM and complications.

We aimed to identify a set of blood-borne miRNAs, which could serve as biomarkers of T2DM and complications in a mixed Israeli Arab/Jewish patient cohort.

## 2. Results

### 2.1. Plasma microRNAs Show Altered Expression between Subject Groups

To characterize changes in miRNA levels that show association with T2DM and microvascular complications of T2DM, 12 candidate miRNAs (miR-16-2-3p, miR-30d, miR-126-5p, miR-4301, miR-183-5p, miR-27b-3p, miR-146a-5p, miR-28-3p, let-7b-5p, miR-486-3p, miR-122-5p, and miR-423) were measured by quantitative reverse transcription and polymerase chain reaction (qRT-PCR) in RNA isolated from plasma of T2DM patients without diagnosed complications (group B, *n* = 29), T2DM patients with diagnosed microvascular complications (group C, *n* = 29), and healthy controls (group A, *n* = 30). All candidate miRNAs except miR-183-5p could be robustly quantified in >90% of plasma-derived RNA samples, with total missing values for 5, 3, 3, 7, 20, 3, 1, 3, 0, 4, 4, and 2 samples for miR-16-2-3p, miR-30d, miR-126-5p, miR-4301, miR-183-5p, miR-27b-3p, miR-146a-5p, miR-28-3p, let-7b-5p, miR-486-3p, miR-122-5p, and miR-423, respectively. Several candidate miRNAs showed significant differential expression between subject groups (Figure 1A). While the changes in the expression of miR-30d and miR-146a-5p (down-regulation and up-regulation, respectively) was associated with the progression of T2DM, several other miRNAs (miR-183-5p, miR-27b-3p, miR-28-3p, miR-486-3p, miR-423) showed opposite changes of expression in the early T2DM and complicated T2DM groups. For miR-28-3p, miR-486-3p, and miR-423, these differences were statistically significant (Figure 1A).

Multiple stepwise regression analysis was employed to assess the ability of miRNA levels to indicate early vs. complicated diabetes mellitus (DM). This resulted in a combination of four miRNAs (miR-146a-5p, miR-16-2-3p, miR-126-5p, and miR-30d) (Appendix A). When incorporating both clinical measurements and plasma miRNA levels, a single miRNA—miR-423—was sufficient to indicate early vs. complicated T2DM with a T value of 3.7, *p* < 0.001 (Appendix A).

### 2.2. Diabetes Mellitus–Associated Changes in Plasma microRNA Levels Differ between Men and Women

Discriminant analysis showed that the plasma levels of miR-146a-5p, miR-16-2-3p, miR-126-5p, miR-30d, and miR-423 (Appendix A) could be used to distinguish between early and complicated T2DM with 76.7% diagnostic accuracy, 79% sensitivity, 75% specificity, 76% positive predictive value, and 22% negative predictive value (Table 1). Repeating the analysis with only the male subjects improved the diagnostic accuracy to 83.3%. Conversely, repeating the analysis with only the female subjects decreased the diagnostic accuracy to 70% (Table 1).

Furthermore, we observed prominent differences between the sex-specific Pearson correlations of specific miRNA levels with clinical measurements (Table 3). Thus, only two of the measured miRNAs showed any significant correlations to any clinical measurements in the entire cohort. However, stratifying the cohort into men and women led to significant observed associations of miRNA levels with multiple anthropometric measurements (Table 3). Specifically, the levels of miR-486-3p showed a strong positive correlation with low-density lipoprotein (LDL) cholesterol levels in women but not in men (Figure 1C), conversely showing a significant negative correlation with high-density lipoprotein (HDL) levels in men but not in women (Figure 1D).

### 2.3. Ethnicity–Associated Differences in Plasma microRNA Levels and Clinical Parameters

To check if plasma miRNA levels or other clinical parameters differ in association with ethnic affiliation, the subject groups were stratified according to ethnic self-identification as Jew, Muslim Arab, Christian Arab, Druse, or undetermined (Appendix A). The Jewish and Muslim Arab groups were chosen for further analysis, as they were the largest two groups in our cohort and included individuals from all 3 clinical categories (A, B, and C). 

Of all miRNAs tested, only miR-183-5p levels differed significantly (*p* = 0.014, *t*-test) between the ethnic groups as a whole, showing higher average levels in the Muslim Arab subjects in all clinical groups, but especially in group B, where its average levels were >3-fold higher (*p* = 0.031, *t*-test) in the Muslim Arabs compared to the Jews (Figure 2A). Additionally, miR-16-2-3p was >2-fold higher (*p* = 0.02, *t*-test) in the Muslim Arabs compared to the Jews in group A (healthy controls) but showed no significant differences in groups B or C (Figure 2A). 

Comparing other clinical measurements, a significant difference was observed between Muslim Arabs and Jews in group C, with height, systolic, and diastolic blood pressure averages all being higher in the Jews. Only in the case of systolic blood pressure (SBP) was there significant (*p* = 0.02) difference between Muslim Arabs and Jews from all three clinical categories (A, B, and C) (Figure 2B).

## 3. Discussion

In this study, we found significant differences between the plasma levels of several miRNAs in subjects with early compared to complicated T2DM, as well as compared to healthy controls, in a mixed Arab/Jewish Israeli population. Several of these miRNAs have been identified previously as potential biomarkers of T2DM or related conditions (miR-30d [18,19], miR-126-5p [18,20], miR-183-5p [21], miR-146a-5p [18,22], miR-28-3p [23,24], miR-122-5p [19,25], and miR-423 [26]), while others have not (miR-16-2-3p, miR-4301, miR-27b-3p, and miR-486-3p). As observed in the case of miR-183-5p and miR-423, the plasma levels of some miRNAs do not always show a linear association with disease progression, but rather a reversal between early and late disease stages. This was shown by our and others’ previous studies, for example [27]. Nor is such a reversal unique to miRNAs; it also occurs with other established pathophysiological hallmarks (for example, kidney output, which may increase in early diabetes and then decline with the accumulation of kidney damage). 

The study was limited by its sample size; additionally, there was a mean age difference between the healthy and diabetic subjects, which could potentially introduce an age-related bias between group A and groups B+C. Age-related changes were assessed in prior studies for some of the plasma miRNAs (e.g., miR-146a-5p, which was found to be unchanged) [28,29]. In all statistical tests used in our study, a sex-related difference was apparent.

While most biomarker discovery studies adjust their statistical analyses for sex (along with other confounding variables), most stop short of looking at women and men as two separate subsets where different biomarkers may be relevant to the same overall condition. This, despite the fact that even the applicability of traditional biomarkers of metabolic disease differs significantly between the sexes, as reported previously [30,31]. Given the well-known physiological and endocrine differences between the sexes, it is not surprising that many miRNAs are differentially expressed and differentially regulated in women and men (reviewed, Reference [32]). For example, in our study, miR-486-3p levels showed significant differences between the healthy and early T2DM groups in men but not women. Furthermore, in women, miR-486-3p levels showed a significant positive association with LDL levels, while in men, a significant negative association with HDL levels was observed. Our findings suggest that T2DM biomarker panels based on circulating miRNAs should be formulated separately for men and women for optimal accuracy and sensitivity, and that the eventual standardization of these panels for clinical practice, while desirable, is not straight forward. 

An additional source of variability that has to be addressed in T2DM biomarker discovery and validation stems from the different genetic/ethnic background of the subjects, as shown previously [20]. In our study, a mixed Arab/Jewish Israeli population was sampled. Although such a group cannot be considered genetically homogenous, neither is it extremely diverse, as history, archeology, and genetics have all indicated a relatively recent common Middle Eastern origin for both groups (see, for example, Reference [33]). We nevertheless observed significant differences in miRNA levels and blood pressure values between Muslim Arabs and Jews in our sample, which should be tested in larger cohorts. Based on the accumulating data, it is altogether feasible that taking ethnic origin, as well as sex, into account will improve the sensitivity and specificity of miRNA-based biomarkers of DM. More studies with larger sample sizes, as well as a re-evaluation of published data are necessary to broaden our understanding of genetic and sex-based effects on circulating miRNA profiles.

Increasing evidence supports endocrine function of circulating miRNAs [34,35]. For example, miR-122-5p, the dominant miRNA in the liver and a known tumor suppressor [36,37], has been shown to be transferred between cells [38] and our and others’ studies have showed that its levels in circulation are altered following bariatric surgery [25,39]. Furthermore, any doubt of the physiological relevance of such signaling by extracellular miRNAs has been put to rest by recent studies [40]. It is therefore imperative to explore the roles of miRNAs and other ncRNAs both as biomarkers and effector molecules in metabolic diseases, roles that can potentially be exploited for prognostic and/or therapeutic purposes. 

## 4. Materials and Methods 

### 4.1. Ethics and Study Participants 

The study was approved by the Research Ethics Committee of the Galilee Medical Center, Nahariya, Israel (0090-15-NHR), and participants gave fully informed oral and written consent. Eighty-eight subjects were recruited during the years 2015–2016 at the Department of Internal Medicine A, the Galilee Medical Center, Nahariya, Israel. The sample size was calculated according to Lee et al. [41] using a calculator available online (http://sph.umd.edu/department/epib/sample-size-and-power-calculations-microarray-studies) and the following assumptions: E(R_0_) = 0.5, G_0_ = 6, power = 0.9, |μ_1_| = 0.6, σ_0_ = 1. The subjects consisted of the following groups:Healthy volunteers (to serve as baseline) with normal response to glucose and insulin and no known renal or vascular pathologies;Early-stage T2DM with no known renal or vascular pathologies; Late-stage T2DM with renal and/or vascular complications including retinopathy or peripheral artery disease (PAD). 

Inclusion criteria (T2DM) included ages 20–80, T2DM with proteinuria, T2DM with retinopathy, T2DM with peripheral vascular disease. Inclusion criteria (healthy) included ages 20–80, no diagnosis of metabolic syndrome, no overweight, no diagnosis of T2DM, no concomitant medications. Exclusion criteria included cancer, dialysis, pregnancy, inability to sign informed consent, and patient choice. T2DM was defined as one of the following: Fasting plasma glucose level ≥7.0 mM (126 mg/dL); plasma glucose ≥11.1 mM (200 mg/dL) two hours after a 75 g oral glucose load (glucose tolerance test); symptoms of high blood sugar and casual plasma glucose ≥11.1 mM (200 mg/dL); or glycated hemoglobin (HbA1C) ≥6.5% as per Diabetes Control and Complications Trial (DCCT) guidelines. Peripheral artery disease was defined as: Nontraumatic limb amputation; limb bypass surgery or percutaneous revascularization; a history of intermittent claudication with ankle-brachial pressure index (ABPI) <0.80 in at least one side; or previous carotid endarterectomy. Retinopathy was defined by ophthalmologic examination such as: Leaking blood vessels; retinal swelling, such as macular edema; pale fatty deposits on the retina (exudates); damaged nerve tissue (neuropathy); and similar changes in the blood vessels. Height, weight, medical historical data, and resting vital signs were recorded at the time of enrollment. Approximately 70% of the subjects were Israeli Arabs (predominantly Muslim), and 25% were Israeli Jews, most of them Sephardi. The distribution of anthropometric measurements and ethnographic data in the participant groups are provided in Appendix A. Ten milliliters of venous blood were collected in standard anticoagulant Ethylenediaminetetraacetic acid (EDTA)-treated vacutainer tubes. All blood samples were centrifuged at 2000 *g* for 10 min to pellet cellular elements immediately after blood draw. To minimize freeze–thaw degradation, the supernatant plasma was aliquoted and immediately frozen at –80 °C. To remove any remaining cellular contents, thawed plasma samples were centrifuged at 15,700 *g* for 10 min, and plasma supernatant was aliquoted into 200 μL volumes for storage at −80 °C for further analysis. 

### 4.2. RNA Isolation and Quantification of Microrna Levels 

Previously, we systematically compared several methods of RNA extraction from plasma [42]. Total RNA isolation used the QIAGEN miRNeasy Serum/Plasma kit (Cat. 217184, QIAGEN, Valencia, CA, USA), with an elution volume of 14 µL nuclease-free water. Samples were lysed in QIAzol lysis reagent (QIAGEN, Cat. 79306). After elution, samples were stored at −80 °C. Reverse transcription and qPCR for candidate miRNAs were performed using TaqMan Advanced miRNA assays (Thermo Fisher Scientific, Waltham, MA USA), with a scaled-down version of the manufacturer’s protocol. Briefly, the polyadenylation of miRNAs was performed in a final volume of 3 µL using 2 µL of RNA; adapter ligation was performed in a final volume of 9 µL; reverse transcription was performed in a final volume of 18 µL; the miR-Amp reaction was performed in a final volume of 30 µL; and qPCR was performed in a final volume of 5 µL, in duplicates, on an Applied Biosystems ABI-7900HT Sequence Detection System equipped with a 384-well block (Thermo Fisher Scientific). 

### 4.3. Choice of Tested microRNAs

Candidate miRNAs were gathered from 11 published studies of miRNAs showing altered levels in the plasma/serum of subjects with diabetes [18,19,20,21,22,23,24,25,26,43,44]. Additionally, we used our own unpublished profiling results. We focused on 12 candidate miRNAs: miR-16-2-3p, miR-30d [18,19], miR-126-5p [18,20], miR-4301, miR-183-5p [21], miR-27b-3p, miR-146a-5p [18,22], miR-28-3p [23,24], let-7b-5p [43,44], miR-486-3p, miR-122-5p [19,25,27], and miR-423 [26].

### 4.4. Statistical Analysis

Results were analyzed with sequence detection system (SDS) 2.3 (Applied Biosystems), Microsoft Excel, WinSTAT, and StatPlus Mac LE (AnalystSoft, Walnut, CA, USA) software. To account for differences in RNA amounts between the individual samples, global normalization of qRT-PCR data was performed, using the median value of relative concentrations for all miRNAs per sample. In our prior studies, e.g., [45], this method has proven to be more robust than relying on specific normalizers. Samples with missing values for a particular miRNA were excluded from downstream analyses involving that miRNA. Student’s *t*-test was used to evaluate differences between groups. Associations between miRNA and clinical variables were assessed using Pearson Correlations, univariate regression and multivariate stepwise regression analyses. Differences were considered statistically significant at *p* < 0.05. Subjects with missing data were excluded from the relevant tests.

## Figures and Tables

**Figure 1 ncrna-05-00032-f001:**
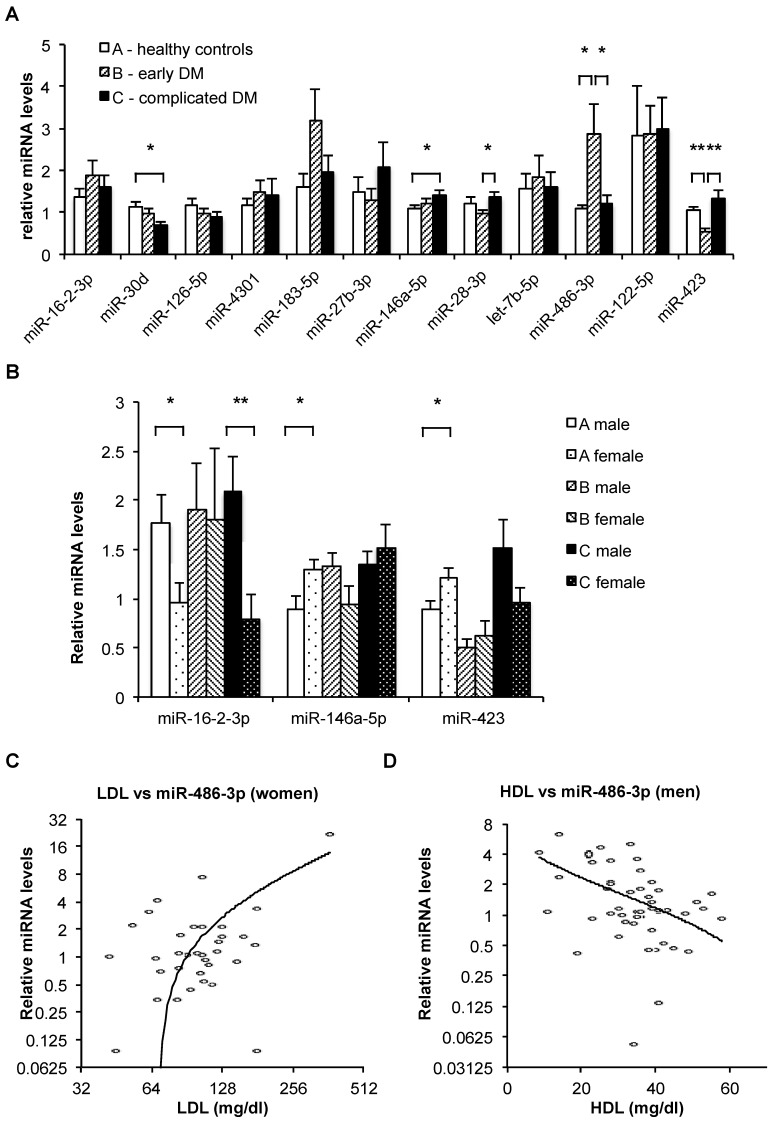
(**A**) Relative mean levels of plasma microRNAs in the three subject groups, as measured by quantitative reverse transcription and polymerase chain reaction (qRT-PCR). The data are normalized (the levels of each miRNA can be compared between groups, but the levels of different miRNAs cannot be compared). Bars, standard error. *N*(A) = 30, *N*(B) = 29, *N*(C) = 29. *: *p* < 0.05; **: *p* < 0.001 (*t*-test). (**B**) Relative mean levels of plasma miR-16-2-3p, miR-146a-5p, and miR-423 in the three subject groups stratified by sex, as measured by qRT-PCR. The data are normalized (the levels of each miRNA can be compared between groups, but the levels of different miRNAs cannot be compared). Bars, standard error. Numbers of samples in each category appear in Table 3. *: *p* < 0.05; **: *p* < 0.01 (*t*-test). (**C,D**) Relative miR-486-3p levels (determined by qRT-PCR) plotted against low-density lipoprotein (LDL) levels in female subjects (**B**, *n* = 36) and against high-density lipoprotein (HDL) levels in male subjects (**C**, *n* = 52). Note logarithmic scale of horizontal axis in B and vertical axis in both B and C, used to improve resolution. Trendlines are logarithmic.

**Figure 2 ncrna-05-00032-f002:**
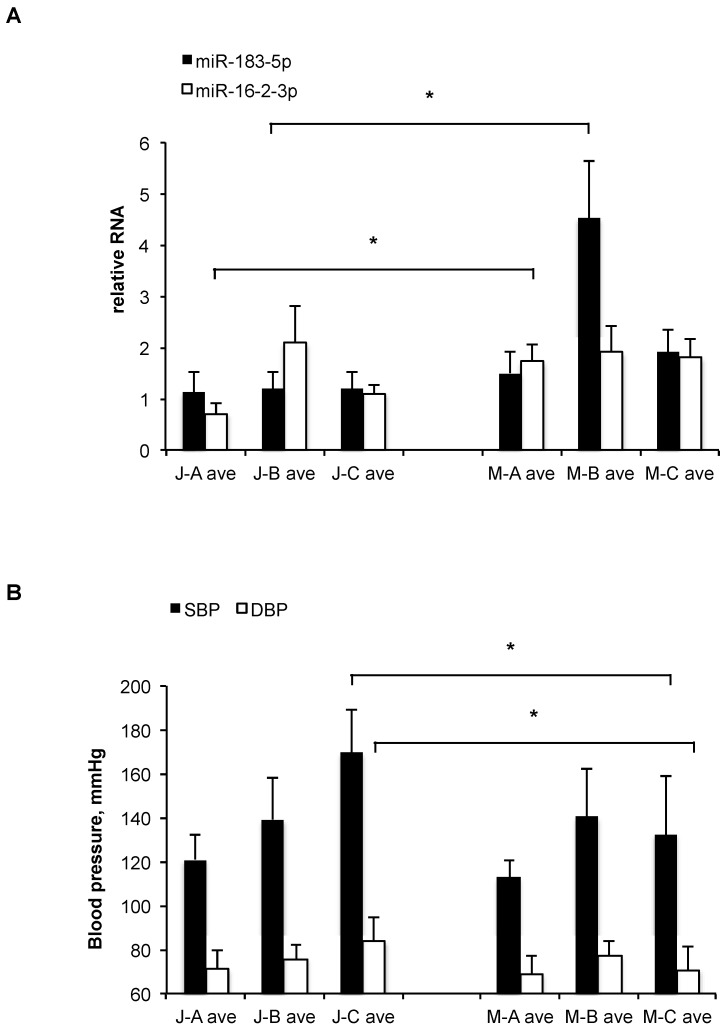
(**A**) Relative mean levels of plasma miR-183-5p and miR-16-2-3p in the three subject groups (A, B, C) with ethnic affiliation of Muslim Arab (M) or Jew (J), as measured by qRT-PCR. The data are normalized (the levels of each miRNA can be compared between groups, but the levels of different miRNAs cannot be compared). Bars, standard error. Numbers of samples in each category appear in Appendix A. *: *p* < 0.05 (*t*-test). (**B**) Mean blood pressure levels (in mmHg) of the three subject groups (A, B, C) with ethnic affiliation of Muslim Arab (M) or Jew (J). SBP, systolic; DBP, diastolic. Bars, standard error. Numbers of samples in each category appear in Appendix A. *: *p* < 0.05 (*t*-test).

**Table 1 ncrna-05-00032-t001:** Discriminant analysis of plasma levels of miR-146a-5p, miR-16-2-3p, miR-126-5p, miR-30d, and miR-423 in distinguishing early vs. complicated diabetes mellitus (DM).

	Actual	Calculated		Total
		Early DM	Complicated DM	
All T2DM samples	Early DM	22	7	29
	Complicated DM	6	21	27
	76.7% of the cases were classified correctly			
DM men only	Early DM	17	1	18
	Complicated DM	5	13	18
	83.3% of the cases were classified correctly			
DM women only	Early DM	7	4	11
	Complicated DM	2	7	9
	70% of the cases were classified correctly			

In the entire cohort (men + women), significant (*p* < 0.05) differences in the plasma levels of miR-30d, miR-146a-5p, miR-28-3p, miR-486-3p, and miR-423 were observed between the groups (Table 2). In addition, miR-183-5p was significantly up-regulated in early type 2 DM (T2DM) (group B) compared to healthy controls (group A), only in female subjects (as opposed to the male subjects and the entire cohort).

**Table 2 ncrna-05-00032-t002:** Summary of significant (*p* < 0.05, *t*-test) differences between plasma miRNA levels in groups of subjects as indicated. Group A: Healthy volunteers; Group B: Early-stage T2DM with no known renal or vascular pathologies; Group C: Late-stage T2DM with renal and/or vascular complications.

	miRNA	Groups	Fold Change in Levels	*p*-Value
Entire cohort (*n* = 88)	miR-30d	A vs. C	0.64	0.003
(group A = 30, group B = 29, group C = 29)	miR-146a-5p	A vs. C	1.30	0.027
	miR-28-3p	B vs. C	1.43	0.029
	miR-486-3p	A vs. B	2.67	0.013
		B vs. C	0.42	0.038
	miR-423	A vs. B	0.52	0.00001
		B vs. C	2.43	0.0003
Women only (*n* = 36)	miR-30d	A vs. C	0.69	0.043
(group A = 15, group B = 11, group C = 10)	miR-183-5p	A vs. B	4.17	0.031
	miR-423	A vs. B	0.51	0.0027
Men only (*n* = 52)	miR-30d	A vs. C	0.59	0.017
(group A = 15, group B = 18, group C = 19)	miR-146a-5p	A vs. B	1.46	0.033
		A vs. C	1.50	0.022
	miR-486-3p	A vs. B	2.19	0.0009
		B vs. C	0.53	0.0013

Within healthy control group A, significant (*p* < 0.05) differences in the plasma levels of miR-16-2-3p, miR-146a-5p, and miR-423 were observed between men and women (Figure 1B). Additionally, a significant (*p* = 0.007) difference in the plasma levels of miR-16-2-3p was observed between women and men in the complicated T2DM group C, while miR-423 showed a similar trend, which did not reach statistical significance (Figure 1B).

**Table 3 ncrna-05-00032-t003:** Summary of significant (*p* < 0.05) Pearson correlations between plasma miRNA levels and clinical measurements as indicated. F—female, M—male. NA refers to the number of subjects in group A, etc.

	miRNA	Measurement	R Value	*p*-Value
entire cohort	miR-183-5p	BMI	0.33	0.048
(*N* = 88; *N*A = 30, *N*B = 29, *N*C = 29)	miR-486-3p	LDL	0.47	0.009
F only	miR-4301	LDL	0.52	0.002
(*N* = 36; *N*A = 15, *N*B = 11, *N*C = 10)	miR-183-5p	WEIGHT	0.38	0.040
		BMI	0.50	0.005
		ABD. GIRTH	0.38	0.037
	miR-486-3p	LDL	0.71	0.000004
	miR-122-5p	DBP	0.45	0.006
		HBA1C	0.43	0.010
	miR-423	AGE	−0.35	0.031
		WEIGHT	−0.53	0.001
		BMI	−0.55	0.0006
		ABD. GIRTH	−0.43	0.007
		GLU. LEVEL	−0.39	0.016
		HBA1C	−0.45	0.006
M only	miR-16-2-3p	HEIGHT	0.39	0.009
(*N* = 52; *N*A = 15, *N*B = 18, *N*C = 19)		SBP	−0.35	0.015
		HDL	−0.34	0.019
	miR-30d	AGE	−0.31	0.032
		SBP	−0.34	0.017
	miR-4301	GLU. LEVEL	0.31	0.035
	miR-27b-3p	WEIGHT	0.45	0.003
		BMI	0.39	0.009
	miR-146a-5p	AGE	0.45	0.001
		BMI	0.37	0.013
		ABD. GIRTH	0.31	0.026
		SBP	0.34	0.016
	miR-486-3p	HEIGHT	0.43	0.004
		HDL	−0.48	0.0005
	miR-122-5p	ABD. GIRTH	−0.28	0.048

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
