# Peer review of "Sex Differences in Plasma MicroRNA Biomarkers of Early and Complicated Diabetes Mellitus in Israeli Arab and Jewish Patients"

_ncrna, 2019, doi:10.3390/ncrna5020032_

Reviewer 1 Report

Findings biomarkers for Type 2 Diabetes mellitus and related complication is a crucial topic; moreover gender medicine research is a developing field, due to the importance of distinguishing between sex in clinical practice.

However for being taken into consideration, I believe the manuscript should be improved and corrected. Several amilioration related to the format ,exposition, and the results explanation are required, as suggested below: 

Major comments:

1-In order to well understand the study, firstly the authors are strongly recommended to re-edit the text writing in an appropiate manner  that allow the readers to well understand the results. To give an example in lines 74-75 it is not clear the list of samples.

2- As you argue in the discussion, the mean age for the control group is not comparable with that  of both early-stage  and late-stage T2DM group. You should use a control group with a comparable mean age.

3-Table 2 is not in a proper format that allow a clear comprehension of the results;by analysing only the table I could not  understand the significance .

4- Figure 1C: if you want to emphasize the correlation, I suggest to  show in the graph the trend line. Furthermore the results showed in figure 1C are not discussed. Why should be of relevance 

these differences between men and women in the correlation of miR-4866-3p and HDL/LDL?

5- Regarding the ethnicity-associated differences in miRNA levels, the compared groups (Jewish and Muslim Arab) are not homogeneous. Even if the data could be interesting, the number of samples in the groups should be increased.

Minor comments:

1- The histograms have incorret labels for y-axes. Please specify that you showed miRNA expression levels in the graph.

2- In table S1, both the ratio and the percentage of sex are redundant information; you can leave the percentage of male or female only.

Author Response

19 March 2019

To: Dr. Clàudia Aunós, Editor

ncRNA

Dear Dr. Aunós,

Thank you for your letter and the referees’ suggestions for the revision of our paper, manuscript ID: ncrna-442907, titled “Sex differences in plasma microRNA biomarkers of early and complicated 
diabetes mellitus in Israeli Arab and Jewish patients”. We have thoroughly revised our manuscript in order to address in detail the referees’ suggestions. This letter addresses the referees’ comments (marked in italics) and details the specific changes made. A Word track changes version of the manuscript is also included as requested. We thank the reviewers for their constructive suggestions and hope that our revised manuscript addresses all their concerns.

Reviewer 1

Findings biomarkers for Type 2 Diabetes mellitus and related complication is a crucial topic; moreover gender medicine research is a developing field, due to the importance of distinguishing between sex in clinical practice.

We thank the reviewer for this acknowledgement of the topic’s importance.

Major comments:

1-In order to well understand the study, firstly the authors are strongly recommended to re-edit the text writing in an appropiate manner  that allow the readers to well understand the results. To give an example in lines 74-75 it is not clear the list of samples.

Lines 74-75 do not list the samples, but rather indicate how many missing values were for each miRNA, in our 88 samples. We have amended this section to more clearly link the missing values with the miRNAs.

2- As you argue in the discussion, the mean age for the control group is not comparable with that of both early-stage and late-stage T2DM group. You should use a control group with a comparable mean age.

As indeed we noted in the Discussion, “there was a mean age difference between the healthy and diabetic subjects, which could potentially introduce an age-related bias between group A and groups B+C.” This was, unfortunately, an external limitation of the study as we had no source for healthy volunteers of matching age to the patient groups. However, age-related changes were assessed in prior studies for some of the plasma miRNAs (e.g. miR-146a-5p); this reference has been added to the Discussion.

3-Table 2 is not in a proper format that allow a clear comprehension of the results; by analysing only the table I could not  understand the significance .

We appreciate the reviewer’s comment and have now revised Table 2 as well as its legend to minimize the use of acronyms and make the results easier to understand.

4- Figure 1C: if you want to emphasize the correlation, I suggest to show in the graph the trend line.

Following the Reviewers’ suggestions, trendlines were added to Fig. 1C-D.

 Furthermore the results showed in figure 1C are not discussed. Why should be of relevance 

these differences between men and women in the correlation of miR-4866-3p and HDL/LDL?

We appreciate the reviewer’s comment and have now added a paragraph to the Discussion addressing these results.

5- Regarding the ethnicity-associated differences in miRNA levels, the compared groups (Jewish and Muslim Arab) are not homogeneous. Even if the data could be interesting, the number of samples in the groups should be increased.

The reviewer is correct in pointing out these objective limitations of the study, as we have also addressed them at length in the Discussion. While additional studies with more samples lie outside the scope of our manuscript, we hope that our findings can instruct such efforts in the future.

Minor comments:

1- The histograms have incorret labels for y-axes. Please specify that you showed miRNA expression levels in the graph.

Figures 1A-D and 2A have been amended as per the Reviewer’s request.

2- In table S1, both the ratio and the percentage of sex are redundant information; you can leave the percentage of male or female only.

Table S1 has been amended as per the Reviewer’s request.

With Kind Regards,

Ari Meerson, PhD

Senior Researcher

Health and Nutrition Dept.

MIGAL

Reviewer 2 Report

In the article entitled "Sex differences in plasma microRNA biomarkers of early and complicated diabetes mellitus in Israeli Arab and Jewish patients" Meerson et. al., identifies plasma miRNAs which can serve as biomarkers of T2DM. Several questions remained unanswered in the current version of the manuscript.

In the qRT-PCR based analysis for differential expression of the miRNAs, the authors have not mentioned in details the normalization strategy, for e.g. what is the reference miRNA that was taken for relative quantification.

For the plasma miRNA differential analysis, certain miRNAs had missing values. How was these missing values were dealt with in the downstream differential analysis?

How does the author explain upregulation of miR-183-5p in early T2DM compared to healthy controls and then find downregulation of the same miRNAs in late T2DM vs early T2DM (Fig1A). For miR-423 they have found the opposite (Fig1A). There is no trend in terms of levels of downregulation/upregulation in healthy vs early and late T2DM.

Although the authors find significant differences in miR-423 in healthy vs early and early vs late T2DM (Fig1A), there is no significant difference between healthy and late T2DM (same is true for miR-183). With these trends in expression it is not justifiable to call these miRNAs as potential biomarkers as they can be random differences, nothing to do with the disease.

Result subsection 2.3, the authors try to detect gender specific differences in miRNA expression which this reviewer believes is important but the result presented seems non-concordant. On what basis were some of the miRNAs represented in a figure form (Fig1B) and the others in a tabular form (Table2)? For e.g. miR-16-2-3p is shown to be significantly different among genders in different groups (A,B or C) but it is absent in Table2.

Where is result subsection 2.2?

The authors found a fold change of "1.3" and "1.43" for miR-146-5p and miR-28-3p respectively in Table2. These values might be statistically significant but are they biologically meaningful?

Fig1C has to be improved and the information that has to be obtained from this plot is not clear.

Line 127, "conversely showing a significant negative correlation with HDL levels in women but not in men (Fig1D)" should be "...HDL levels in men but not in women (Fig1D)".

Results subsection 2.4 is an interesting and equally important source of variability that the authors have tried to address in terms of ethnicity in T2DM but the number of samples are way too less for claiming a miRNA as a biomarker.

Referencing has to be better.

Author Response

19 March 2019

To: Dr. Clàudia Aunós, Editor

ncRNA

Dear Dr. Aunós,

Thank you for your letter and the referees’ suggestions for the revision of our paper, manuscript ID: ncrna-442907, titled “Sex differences in plasma microRNA biomarkers of early and complicated diabetes mellitus in Israeli Arab and Jewish patients”. We have thoroughly revised our manuscript in order to address in detail the referees’ suggestions. This letter addresses the referees’ comments (marked in italics) and details the specific changes made. A Word track changes version of the manuscript is also included as requested. We thank the reviewers for their constructive suggestions and hope that our revised manuscript addresses all their concerns.

Reviewer 2

In the qRT-PCR based analysis for differential expression of the miRNAs, the authors have not mentioned in details the normalization strategy, for e.g. what is the reference miRNA that was taken for relative quantification.

In fact, we have described the normalization strategy in Section 4.4: “To account for differences in RNA amounts between the individual samples, global normalization of qRT-PCR data was performed, using the median value of relative concentrations for all miRNAs per sample.” In our prior studies this method has proven to be more robust than relying on specific normalizers. This information has been added to the Methods (section 4.4).

For the plasma miRNA differential analysis, certain miRNAs had missing values. How was these missing values were dealt with in the downstream differential analysis?

Samples with missing values for a particular miRNA, were excluded from downstream analyses involving that miRNA.  This information has been added to the Methods (section 4.4).

How does the author explain upregulation of miR-183-5p in early T2DM compared to healthy controls and then find downregulation of the same miRNAs in late T2DM vs early T2DM (Fig1A). For miR-423 they have found the opposite (Fig1A). There is no trend in terms of levels of downregulation/upregulation in healthy vs early and late T2DM. Although the authors find significant differences in miR-423 in healthy vs early and early vs late T2DM (Fig1A), there is no significant difference between healthy and late T2DM (same is true for miR-183). With these trends in expression it is not justifiable to call these miRNAs as potential biomarkers as they can be random differences, nothing to do with the disease.

The reviewer is correct in noting that the plasma levels of some miRNAs do not always show a linear association with disease progression, but rather a reversal between early and late disease stages, as shown by our and others’ previous studies (reference added). This is not unique to miRNAs; it also occurs with other established pathophysiological hallmarks (for example, kidney output which may increase in early diabetes and then decline with the accumulation of kidney damage). This point was added to the Discussion.

Result subsection 2.3, the authors try to detect gender specific differences in miRNA expression which this reviewer believes is important but the result presented seems non-concordant. On what basis were some of the miRNAs represented in a figure form (Fig1B) and the others in a tabular form (Table2)? For e.g. miR-16-2-3p is shown to be significantly different among genders in different groups (A,B or C) but it is absent in Table2.

The figures and tables focus on different findings in the data. Thus, Fig. 1A presents overall significant differences between subject groups A, B and C without stratifying them by sex. Fig. 1B presents significant differences between men and women only, within each group, and only for those miRNAs where such differences were observed. Table 2 presents significant differences between the subject groups but also after stratification by sex. In this case a tabular format was chosen because the number of data points and significant differences would make a bar chart too crowded. miR-16-2-3p levels showed significant differences between men and women within groups A and C, thus it is presented in Fig. 1B; however we did not observe significant differences between its levels in the subject groups even when stratified by sex, therefore it is not presented in Table 2.

Where is result subsection 2.2?

We thank the reviewer for noticing the error in the numbering of sections and have now corrected it.

The authors found a fold change of "1.3" and "1.43" for miR-146-5p and miR-28-3p respectively in Table2. These values might be statistically significant but are they biologically meaningful?

There is abundant evidence that robust 30%-40% changes in the levels of particular miRNAs (or in gene expression in general) can have a biological effect, especially in the context of chronic disease, when there is a cumulative exposure over months and indeed years.  However, our current study only concerns itself with miRNA-based biomarkers, and not with the biological effects of the altered miRNA repertoire.

Fig1C has to be improved and the information that has to be obtained from this plot is not clear.

Following the Reviewers’ suggestions, trendlines were added to Fig. 1C-D.

Line 127, "conversely showing a significant negative correlation with HDL levels in women but not in men (Fig1D)" should be "...HDL levels in men but not in women (Fig1D)".

We thank the reviewer for noticing this error and have now corrected it (currently in lines 150-1).

Results subsection 2.4 is an interesting and equally important source of variability that the authors have tried to address in terms of ethnicity in T2DM but the number of samples are way too less for claiming a miRNA as a biomarker.

The reviewer is correct in pointing out these objective limitations of the study, as we have also addressed them at length in the Discussion. While additional studies with more samples lie outside the scope of our manuscript, we hope that our findings can instruct such efforts in the future.

Referencing has to be better.

We are not certain about the meaning of the Reviewer’s last comment, but would be happy to include or correct any pertinent reference that we may have missed.

With Kind Regards,

Ari Meerson, PhD

Senior Researcher

Health and Nutrition Dept.

MIGAL

Round  2

Reviewer 1 Report

Overall, the article has been revised properly.

Authors answered all the points in an exhaustive manner.

Author Response

We thank the Reviewer for the helpful comments and the favourable assessment.

Reviewer 2 Report

Reference at some places are missing for example the first paragraph of introduction does not have a reference. 

Author Response

Dear Drs., 

To address the Reviewer's comment, 2 additional references (to current general reviews on the epidemiology and pathology of type 2 diabetes) were included in the first paragraph of the Introduction. We hope that these are adequate to introduce the subject.

Sincerely, 

Dr. Ari Meerson